## Original Research Article

3D sepal morphology; mutant analysis; size and shape robustness.

**Corresponding author:**
Françoise Monéger.
Email: francoise.moneger@ens-lyon.fr

**Associate Editor:**
Ali Ferjani

# A 3D morpho-space of sepal geometry reveals the importance of organ curvature

Virginie Battu[1], Annamaria Kiss[1], Abigail Delgado-Vaquera[1], Fabien Sénéchal[2], Corentin Mollier[1,3], Diego A. Hartasánchez[1,4], Arezki Boudaoud[1,5] and Françoise Monéger[1]

[1]Laboratoire Reproduction et Développement des Plantes, Université de Lyon, ENS de Lyon, CNRS, INRAE, UCBL, Lyon, France; [2]UMRT INRAE 1158 BioEcoAgro—BIOPI Biologie des Plantes et Innovation, Université de Picardie, Amiens, France; [3]Laboratoire de Génétique et Biologie du développement, Institut Curie, Université PSL, Paris, France; [4]Department of Computational Biology, University of Lausanne, Lausanne, Switzerland; [5]LadHyX, CNRS, École Polytechnique, Institut Polytechnique de Paris, Palaiseau, France

## Abstract

How robust three-dimension (3D) organ shape emerges during morphogenesis is a fundamental question in biology. Addressing this question requires a comprehensive quantification of organ geometry in 3D. To tackle these issues, we considered the sepal of Arabidopsis as a model. Using a unique pipeline allowing to recover 3D sepal morphology, we analysed fifteen mutants affected in different pathways. The results of a Principal Component Analysis reveal sepal curvature as an important parameter accounting for variations in sepal morphology within genotypes. Unexpectedly, despite genetic homogeneity of the wild-type plants and reproducible culture conditions, we found a significant level of variability in sepal morphology. Our data also show that sepal shape from wild-type plants is more robust (less variable) than sepal size, hinting to a possible selective pressure on shape parameters.

## 1. Introduction

Flowers have been extensively used by taxonomists to describe and classify plant species (Linné, 1737). For many decades, the molecular basis of flower development has been under focus, and numerous genetic screens have been performed to isolate mutants with altered flower organ identity or morphology. The functions of many proteins in flower development have been unravelled thanks to the drastic phenotypes identified in such screens (reviewed in Heisler et al., 2022; Hidalgo et al., 2022; Xu et al., 2021). Nonetheless, many genetic mutations produce subtle or hardly visible phenotypes, making it difficult to conclude the biological role of the encoded protein.

An important aspect of a phenotype concerns the extent of its reproducibility. Phenotypic robustness is defined as the reproducibility of a given phenotypic upon a perturbation (Félix & Barkoulas, 2015; Hong et al., 2018). Perturbations can originate from the environment or from the organism itself, for example, following genetic mutation or due to stochastic gene expression (Araújo et al., 2017). In a developmental context, the morphology of organs or organisms is defined by quantitative traits such as size and shape, the values of which can vary to different degrees following a given perturbation: a robust trait will not vary much, whereas a sensitive trait will vary a lot. It is interesting to identify the role of particular genes in this sensitivity. One approach is to analyse mutants and see if a trait of interest is more or less variable in the mutant compared to the wild type in a given environment.

It is, therefore, essential to have a quantitative framework to precisely describe organ geometry to analyse mutants with perturbed organ morphology or organs from wild-type plants grown in different environmental conditions and to study the range of phenotypic variation. Such a framework would allow the study of the relationship between size and shape, for instance, as well as the robustness of geometrical parameters upon a given perturbation.

**Table 1. List of genes selected for mutant and phenotypic analysis.** For each gene, we show its AGI locus code, allelic details for the mutants analysed, genetic background, gene category and reference providing mutant information (Bowman & Smyth, 1999; Kohorn et al., 2006; Pekker et al., 2005; Ravet et al., 2009; Roeder et al., 2010, 2012).

| GENE | AGI | Mutant allele | Background | GENE CATEGORY | Mutant information |
|---|---|---|---|---|---|
| TIP2 | At4g17340 | SALK_131664 | Col-0 | WATER TRANSPORT | Figure S1 |
| PME32 | At3g43270 | pme32-2 | Col-0 | PRIMARY CELL WALL | Figure S1 |
| PMEI3 | At5g20740 | SALK_021135 | Col-0 | PRIMARY CELL WALL | Figure S1 |
| UGD3 | At5g15490 | SALK_006233 | Col-0 | PRIMARY CELL WALL | Figure S1 |
| WAK1 | At1g21250 | wak1-1 | Col-0 | PRIMARY CELL WALL | Figure S1 |
| WAK2 | At1g21270 | wak2-1 | Col-0 | PRIMARY CELL WALL | Kohorn et al., 2006 |
| KNAT7 | At1g62990 | SALK_002098C | Col-0 | SECONDARY CELL WALL | Figure S1 |
| VND4 | At1g12260 | SALK_058195C | Col-0 | SECONDARY CELL WALL | Figure S1 |
| ETT | At2g33860 | ett-22 | Col-0 | AUXIN | Pekker et al., 2005 |
| FER3 | At3g56090 | fer3 | Col-0 | OXYDATIVE STRESS | Ravet et al., 2009 |
| FER1-3-4 | At5g01600 At3g56090 At2g40300 | fer1-3-4 | Col-0 | OXYDATIVE STRESS | Ravet et al., 2009 |
| STO | At4g12970 | GABI_411D10.1 | Col-0 | CELL TYPE | Figure S1 |
| LGO | At3g10525 | lgo-1 | Ler | CELL TYPE | Roeder et al., 2010 |
| KRP1-OE | At2g23430 | pATML1:KRP1 | Ler | CELL TYPE | Roeder et al., 2012 |
| CRC | At1g69180 | crc-1 | Ler | GYNOECIUM | Bowman and Smyth, 1999 |

We use the sepal of Arabidopsis as a model to study determinants of organ morphology. It is the most external floral organ and has many advantages, such as a large number of them on each inflorescence, being easily accessible for dissection, imaging and experimentation, and very reproducible morphology (Roeder, 2021). In a previous work, we developed a semi-high-throughput pipeline allowing quantitative recovery of three-dimensional (3D) geometry from individual sepals and analysed both morphological parameters and transcriptome from 27 individual sepals. We identified five modules of genes whose expression correlated significantly with sepal morphology, revealing biologically relevant gene regulatory networks (Hartasánchez et al., 2023). Here, we took advantage of this 3D pipeline to evaluate the effect of mutating genes involved in different pathways expected to affect sepal morphology. We generated data on 3D sepal morphology from 15 mutants impaired in various cellular processes (see Table 1) and analysed the data with appropriate statistical methods. The purpose was not only to see if mutations affect sepal morphology but also to establish a quantitative framework to analyse 3D morphology and phenotypic variability. The results reveal the importance of sepal curvature in variations of morphology and different variability levels according to shape and size parameters.

## 2. Methods

### 2.1. Plant material

Plants were grown on potting soil in growth chambers at 20°C in short-day conditions (8 h light/16 h darkness) for 20 days before being transferred to long-day conditions (16 h light/8 h darkness) at 22°C. We selected 15 mutants in Col-0 and Ler backgrounds (Table 1).

### 2.2. Molecular characterisation of the mutants tip2, pme32, pmei3, ugd3, wak1, knat7, vnd4 and sto

To characterise T-DNA insertion in the mutants, we extracted genomic DNA from young leaf tissue according to Edwards et al., (1991) without isopropanol precipitation. PCR-genotyping was performed using GoTaq (Promega) following the manufacturer's instructions. PCR products were sent for sequencing to GATC Biotech or Microsynth to precisely locate the T-DNA insertion site. We then selected mutants homozygous for the insertion to perform the analyses. Apart from the *vnd4* mutant, which exhibits a higher expression of the *VND4* gene, all the others are likely loss-of-function mutants (Figure S1).

### 2.3. RNA extraction and qRT-PCR

Total RNA was extracted from inflorescences using Spectrum Plant Total RNA Kit (Sigma). RNA was then treated with DNase (On-Column DNase I Digestion Set from Sigma) to eliminate DNA contamination, and 1.5 µg were subsequently reverse transcribed using RevertAid Reverse Transcriptase enzyme (Thermo Fisher Scientific). Five µl of 1/20 diluted cDNA was subjected to qPCR using SYBR Green (Roche) in a Step One Plus machine (Applied). The efficiency of each primer pair was determined by standard curves using serial dilutions of PCR product. PCR was performed using a three-step protocol with a melting curve. Results were normalised to the expression of either the GAPDH gene (At3g04120) or the TCTP gene (At3g16640), which were chosen using BESTKEEPER

(Pfaffl et al., 2004) and analysed following the method described by (Pfaffl, 2001). Each expression value was determined from the mean of 4 to 5 plants.

### 2.4. Sepal growth curves

For each genotype, we measured abaxial sepal curvilinear length from all the flowers of an inflorescence to make sure that sepals from open flowers (stage 13 according to Smyth et al., 1990) have reached their final size (Figure S2).

### 2.5. Sepal shape and size analysis

We focused on the abaxial sepal, the first of the four sepals to emerge and always located on the external side relative to the axis of the inflorescence. Abaxial sepals from open flowers were detached and incubated for 1h in water containing propidium iodide (PI) at 67 μg/ml. Sepals were then transferred to 0.8% agarose plates in water to avoid dehydration and imaged under an LSM700 confocal microscope equipped with a 5X objective using excitation with laser 555nm and emission ranging from 555 to 630nm. Segmentation and extraction of geometrical parameters were performed as described in Hartasánchez et al. (2023). We measured curvilinear length (Length), curvilinear width (Width), area (Area; independently calculated), aspect ratio (AspRatio; Length divided by Width) and the square root of Gaussian curvature (Curv; the square root of the product of transversal curvature and longitudinal curvature). Due to limited z resolution, sepal thickness was not taken into account.

### 2.6. Statistical analyses

We systematically compared the mutant samples with their corresponding wild-type controls. To do that, we started with an Agostino-Pearson normality test. As not all samples had a normal distribution, we continued by using non-parametric tests. Namely, for each measured parameter, we used the Kolmogorov-Smirnov test for comparing distributions and Bartlett's test for comparing variabilities (standard deviation divided by the mean) and assessed the significance of the difference between mutant and corresponding wild-type samples. In parallel, we considered the available set of wild-type samples and computed the inter and intra-experiment variability. Then, in order to construct a common space in which we could represent all the samples at the same time, we performed a PCA on the mutant samples and projected the wild-type samples into the space defined by the first principal components.

## 3. Results

### 3.1. Rationale for the choice of mutants

Our recent paper (Hartasánchez et al., 2023) highlights the existence of modules of genes that are co-expressed, likely working together, and which strongly correlate with sepal morphology. In continuity with that study, we chose to analyse mutants for seven genes identified as potentially relevant in sepal morphology: *TIP2, UGD3, KNAT7, VND4, FER1, FER3* and *FER4* (see Figure S1 for details). We analysed mutants altered in different biological pathways known to be important in plant development, such as water transport (*tip2*), primary (*pme32, pmei3, ugd3, wak1, wak2*) and secondary (*knat7, vnd4*) cell wall, auxin signalling (*ett*), oxidative stress (*fer1, and fer1 fer3 fer4 triple mutant*) or epidermal cell types

(*sto, lgo, KRP OE1*). In addition, we wanted to evaluate the influence of gynoecium on sepal morphology (using the *crc* mutant), the two organs being in close contact during flower development (see Table 1 and Figure S1 for details).

### 3.2. Quantification of sepal morphology

Based on the above considerations, we collected data for 15 mutants (see Table 1). In order to compare the size and shape of the sepals from different genotypes, we only analysed fully grown sepals to overcome the possible effect of a mutation on sepal growth duration or dynamics (see Methods). For each genotype, we quantified the 3D morphology of 45 sepals taken from 3 different plants and compared it to the corresponding wild-type plants which have been grown at the same time (in some experiments, two batches of mutant plants were compared to a single batch of wild-type plants grown together). Figure 1 shows representative sepals for all the genotypes analysed based on the PCA analysis (see below) (we chose the sepal closest to the mean of the samples). For each sepal, we recovered five 3D descriptors: length, width and area for the size, and aspect ratio and curvature for the shape (see Methods).

### 3.3. Effects of mutations on sepal morphology

We compared the morphology of the mutant sepals with respect to their corresponding wild types (Table 2 and Figure S3). Among the 15 mutants, four of them do not show a difference in any parameter compared with their wild types: *pmei3, ugd3, wak1* and *knat7*. Two of them exhibit a difference in all five parameters compared with their wild types: *fer3* and *KRP1-OE*. All the other mutants have at least one parameter which differs from the wild type. Some mutations affect a given parameter in a positive manner (e.g., *tip2* mutants have longer sepals compared to wild type), while some mutations have a negative effect (e.g., *ett* mutants have shorter sepals compared to wild type). Some mutations have a positive effect on one parameter and a negative effect on another one (e.g., *tip2* mutants have sepals which are longer but narrower compared to wild type). We summarised the mutants' phenotypes to visualise the effect of the genes on sepal size (Figure 2).

### 3.4. Correlations among parameters and principal component analysis

We analysed the correlation between each of the five morphological parameters both in wild-type and mutant samples. First, in order to characterise correlations in the wild type, we considered a large sample, where we pooled all available wild-type samples (Figure 3A).

In wild type, as expected, area is strongly correlated with length and width. Length and width are also positively correlated despite length and width being variable. Interestingly, length does not correlate with aspect ratio, but we observe anti-correlation between curvature and length or area, showing that longer and larger sepals tend to be less curved. In parallel, we looked at correlation heat maps in each wild-type experiment separately (see Supplementary Figure S5). Individual wild-type experiments contain less sepals than pooled wild-type experiments, and some correlation values are not significant there (values without an asterisk on the heat map). Although the set of this correlation heat-maps looks heterogeneous, one may notice that the correlation among size parameters (length, width, area) is significant in most wild-type samples, while the anti-correlation between curvature and length described above

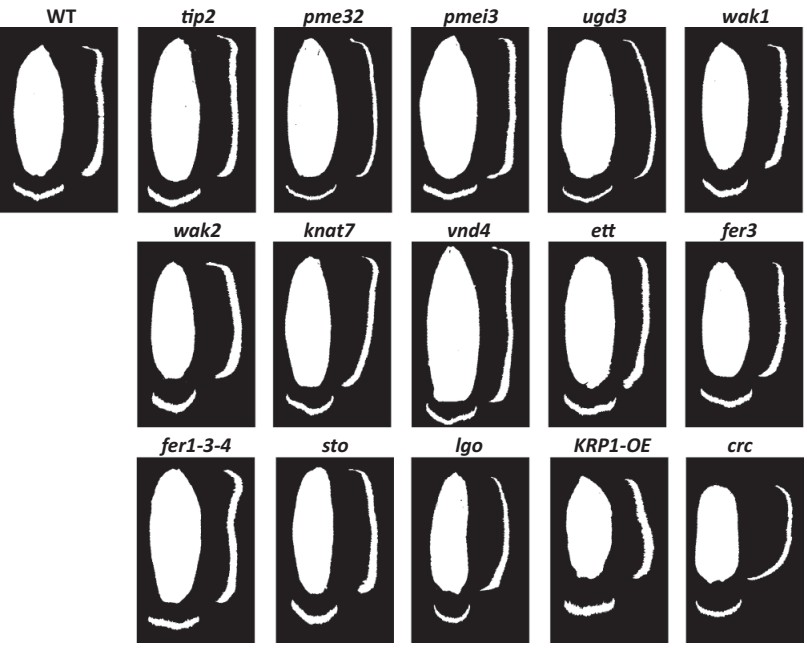

**Figure 1. Representative sepals for each mutant and all the wild types.** We used a Principal Component Analysis (PCA) to select the most central sepal for each genotype in the PCA space. For each representative sepal, we show the top view together with orthogonal views. WT corresponds to a sepal from a Col-0 background, which is the most representative of all the wild-type samples, including the Ler genotype.

**Table 2. Kolmogorov-Smirnov test significance was obtained for each parameter and each mutant compared to its corresponding wild type.** Mean values smaller in the mutant compared to its wild type are highlighted in red, and values higher in the mutant compared to its wild type are highlighted in green. ns indicates that the difference between the distributions of the mutant and wild-type samples is not significant (p-value>0.05), while significance is indicated with asterisks: * if 0.005<p-value≤0.05, ** if 0.0005<p-value≤0.005, *** if 0.00005<p-value≤0.0005 and **** if p-value≤0.00005.

| Genotype | Length | Width | Area | AspRatio | Curv |
|---|---|---|---|---|---|
| tip2 | **** | * | ns | **** | ns |
| pme32 | ** | ns | * | ns | ns |
| pmei3 | ns | ns | ns | ns | ns |
| ugd3 | ns | ns | ns | ns | ns |
| wak1 | ns | ns | ns | ns | ns |
| wak2 | ns | * | ns | ns | ns |
| knat7 | ns | ns | ns | ns | ns |
| vnd4 | * | *** | ns | *** | ns |
| ett22 | **** | ns | *** | ** | ** |
| fer3 | * | **** | **** | * | * |
| fer1-3-4 | * | *** | * | * | ns |
| sto | ** | ns | * | ** | ns |
| lgo | ** | * | * | ns | ns |
| KRP1-OE | **** | * | ** | **** | **** |
| crc | **** | ns | ** | **** | * |

| | Pmut < Pwt | Pmut > Pwt |
|---|---|---|

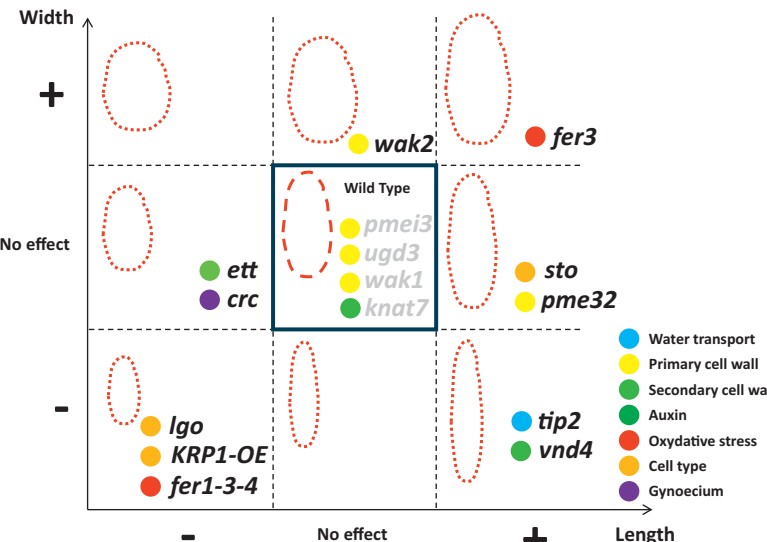

**Figure 2. Effects of the mutations on sepal size (length and width).** The mutants are colour-coded according to their gene category, as shown on the right. Mutants with a phenotype are shown in black font; mutants without a phenotype (i.e. similar to wild type) in grey font.

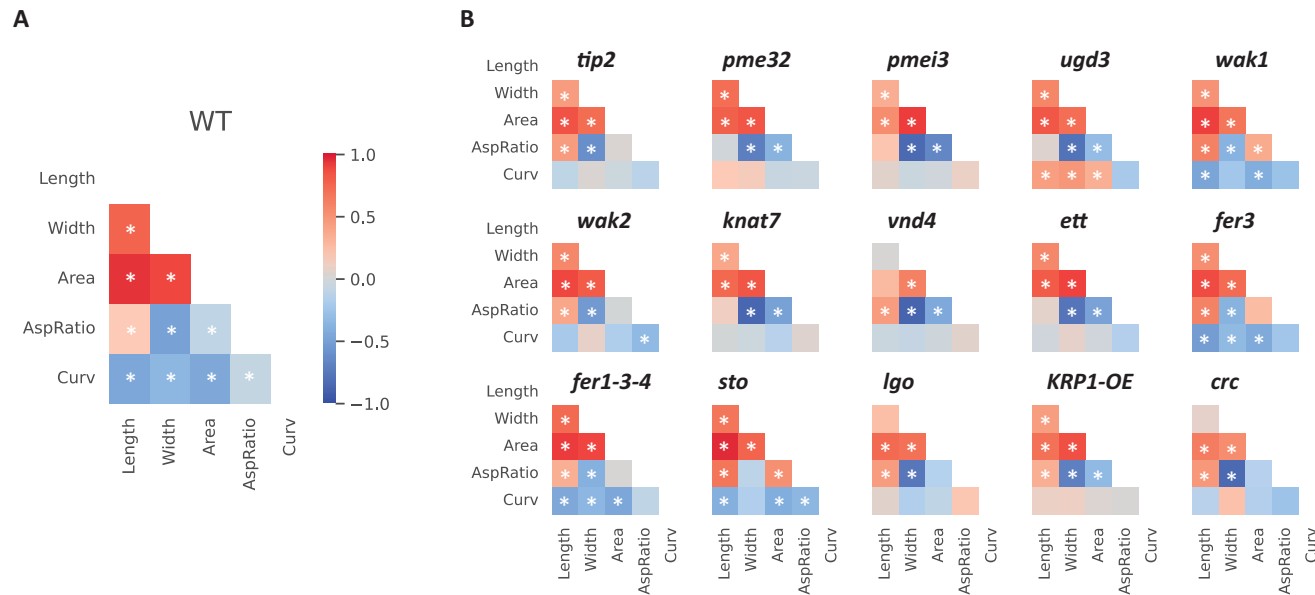

**Figure 3. Correlation heat map of morphological parameters for (A) wild-type (considering the pool of all wild-type samples) and (B) mutant samples.** The common colour bar represents Pearson correlation coefficients. Significant correlations are indicated with an asterisk (∗ if p-value>0.05 when testing for no correlation).

is less obvious separately in each experiment. We wanted to see whether this anti-correlation between curvature and size is clearer when the set of different experiments is taken together. Therefore, in Supplementary Figure S6, we considered the collection of wild-type samples, in which each sample is described by the mean and variability of the five parameters. Although this collection of samples is not large (N=12), we see that the anti-correlation between the curvature and length is indeed significant, while the correlation among size parameters is also clear. This is in accordance with previous findings (Hartasánchez et al., 2023) showing that larger (meaning higher area) sepals tend to be flatter (less curved).

We then analysed the mutant samples (Figure 3B). The correlation heat maps of the mutants seem to be quite similar at first sight. However, some of the mutants appear to differ from the wild type. A weaker correlation between area and length, for instance,

can be seen for *vnd4*. Length and width seem to be completely uncorrelated in *vnd4* and *crc*. For some mutants, like *pmei3*, *knat7* and *vnd4*, width shows a higher anti-correlation strength with aspect ratio and a lower one with curvature. We then performed a PCA to evaluate the relative contribution of each morphological parameter. We first normalised all parameters by their mean in the pooled wild-type samples. Then, to construct a common vector space in which we can represent all the samples without giving too much weight to the wild-type samples, we performed the PCA only considering the mutant data, and we projected the wild-type samples in the space defined by the principal components. By doing so, we implicitly assumed that mutants have more degrees of freedom for the variations of morphological parameters. Figure 4A shows that the first principal component (PC1) is highly correlated with curvature and explains ~75% of the variance. The second

principal component (PC2), which explains slightly less than 20% of the variance, is based mostly on sepal size (length, width and area). The third principal component (PC3), explaining slightly less than 5% of the variance, accounts mostly for the aspect ratio. Wild-type samples show considerable variability along the PC1–PC2 space (Figure 4B). Note that, taken together, wild-type samples show a clear anti-correlation between curvature and size, basically falling along a diagonal line in the PC1–PC2 space, in line with the anticorrelations between individual parameters mentioned above. If mutants are plotted on the same PC1-PC2 space (Figure 4C left), we also observe most mutants falling along the diagonal. However, for the mutant phenotype to be adequately evaluated, every mutant line was evaluated with respect to its corresponding wild type (Supplementary Figure S4). Accordingly, and in order to have a global overview of the relative position of the different mutants with respect to their wild type in the same principal component space, we translated each experiment (pair of wild type and mutant) such that the wild type has its centre in the origin (Figure 4D). The repositioned mutant samples allow for an overview of the effects of the mutated genes on the mean values of geometrical parameters.

### 3.5. Phenotypic variability

Next, we focused on variability, defined for each parameter as its standard deviation normalised by its mean. First, we computed the intra and inter-experiment variabilities of all wild-type samples (Figure 5). Interestingly, the inter-experiment variability (horizontal black lines) of shape parameters (aspect ratio and curvature) is below all the intra-experiment variabilities (black and grey bars), while the parameters related to size (length, width, area) do not have this property. This shows that for the wild type, sepal shape is less variable than sepal size when experiments are repeated. We then compared mutant and wild-type variabilities by performing a Bartlett test to assess the significance of the differences (Figure 5). Most of the parameters show similar variability in mutants and corresponding wild types. However, variability is increased in some mutants compared to the wild type: *vnd4* for width, *ett22* for length, area and aspect ratio, *sto* for aspect ratio, *lgo* for aspect ratio and curvature, *KRP1-OE* and *crc* for curvature. Inversely, variability is reduced in some other mutants compared to the wild type: *tip2* for length and area, *pme32* for aspect ratio, *pmei3* for length and area, *sto* for width, *lgo* for are*a* and *crc* for length. It is interesting to note that the *pmei3* mutant did not exhibit any difference from the wild type in terms of mean shape and size, but it has a more reproducible size (less variable length and area). Lastly, curvature is the most variable of the five parameters. This result is consistent with the PCA revealing sepal curvature as an important parameter accounting for variation in sepal morphology within genotypes.

## 4. Discussion

In this work, we took advantage of a pipeline allowing quantitative recovery of 3D geometry from individual sepals (Hartasánchez et al., 2023) to revisit mutants, try and detect phenotypes even subtly and evaluate phenotypic variability. Interestingly, four of the six mutants selected from modules of co-expression exhibit a phenotype of sepal morphology compared to the wild type, showing that the detection of gene modules and important genes within such modules is a valuable resource. To analyse the data without a preconceived idea of which parameter should be considered and to reveal natural relationships, we performed a PCA analysis. Following the same idea, Bensmihen et al. (2008) performed an allometric

analysis of leaves from a collection of mutants. They showed that leaf width contributes importantly to one of the main principal components in both Arabidopsis and Antirrhinum, suggesting that control of leaf width may be under selective pressure (Bensmihen et al., 2008). Although their analysis included a high number of mutants in two different plant species, it was performed in two dimensions. Interestingly, our three-dimensional analysis of sepals shows that curvature is the parameter which explains most of the variation in sepal shape, highlighting the importance of looking at 3D data when studying organ morphology. Compared to the leaf, our study on the sepal points to a more complex situation as the curvature has the main contribution to the first principal component, followed by size (length, width and area). These results are consistent with the hypothesis that selective pressure acts on sepal curvature and size, resulting in efficient protection of the inner organs of the flower. It would be interesting to see if similar conclusions also hold for other flowering plant species.

### 4.1. Effect on sepal size

Our analysis of mutant phenotypes shows that some genes contribute to sepal size either positively (*KRP1*, *LGO* and the three genes *FER1*, *FER3* and *FER4* when mutated together) or negatively (*FER3*). Giant cells are very abundant on the abaxial epidermis of sepals. We find that the sepals from the *lgo* mutant, which have no giant cells (Roeder et al., 2010), are shorter and narrower compared to the wild type, suggesting that LGO promotes increased sepal size. In an attempt to explore the opposite function, we also analysed plants over-expressing KRP1, whose sepals are covered with giant cells (Roeder et al., 2010), and found that sepals from these plants are also shorter and narrower compared to the wild type, suggesting that too many giant cells might interfere with sepal growth. These results suggest that an appropriate balance between the different cell types and cell sizes in the epidermis is required for proper sepal morphology. This result is in accordance with conclusions from previous work showing the importance of epidermal patterning during sepal development in Arabidopsis (Roeder et al., 2010). In addition, we note that the *KRP1-OE* plants exhibit a drastic change in sepal shape since both the aspect ratio and the curvature are reduced.

In our previous study, we found that the expression of each of the three *FERRITIN* genes positively correlates with the two others as well as with sepal width, length and area (Hartasánchez et al., 2023). Accordingly, the triple mutant *fer1-3-4* is affected in size, and particularly in width, in a negative manner. Surprisingly, the *fer3* mutant has increased sepal size, in particular, increased width. We can interpret these results as the fact that when only one gene is mutated (*fer3*), the plant can compensate and even over-compensate as the size of the sepal is larger than the wild type. However, when three genes are lost, the system is not able to compensate, and we observe a clear reduction in sepal growth in the triple mutant. This observation is consistent with the *fer1-3-4* mutant having higher levels of reactive oxygen species and decreased leaf growth (Ravet et al., 2009).

### 4.2. Effect on sepal shape

Other genes contribute more to sepal shape, affecting length and width in a different manner (*TIP2*, *PME32*, *WAK2*, *VND4*, *ETT*, *STO2* and *CRC*). Mutants in *WAK2* exhibit wider sepals, suggesting that the gene has a negative effect on sepal width. Plant cell walls are embedded in a pectin matrix, which is physically linked with

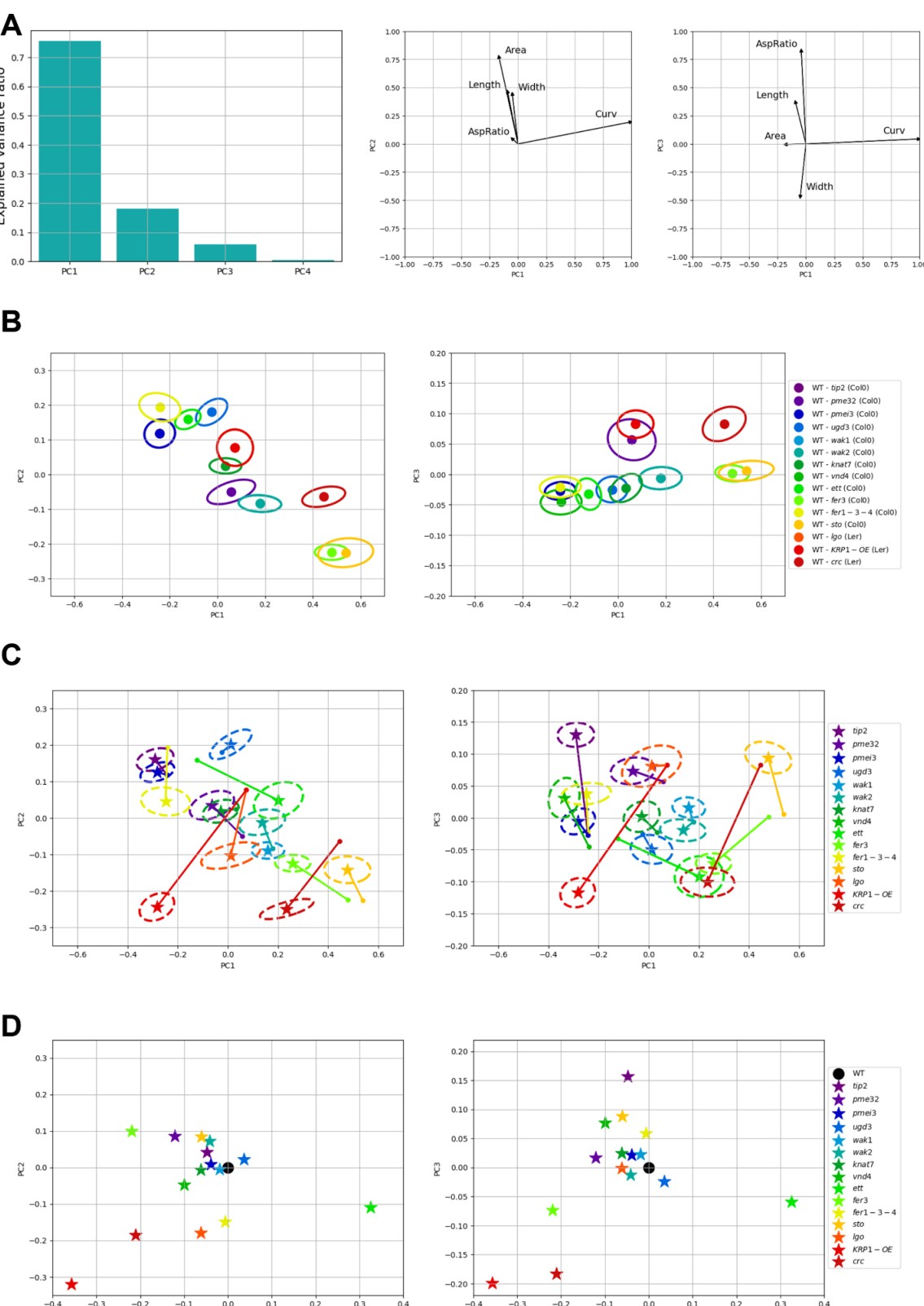

**Figure 4. Wild-type and mutant samples in the PCA space.** (A) Characterisation of the principal components by the explained variance ratio for the first four principal components and the projection of the measured parameters in the principal space defined by the first three principal axes (principal planes PC1-PC2 and PC1-PC3). (B) Wild-type samples projected in the principal planes. Confidence ellipses are centred on the mean marked by a dot, and the ellipse semi-axis lengths are set to 20% of the standard deviation. Each mutant (and its corresponding wild type) has been assigned an arbitrary colour. (C) Mutant samples in the principal component space. Confidence ellipses are defined as in B. The mean mutant (star) is connected to the corresponding mean wild type (point) by a straight line. (D) Mutant and wild-type pairs are shifted in the principal space so that each wild-type sample is located at the origin.

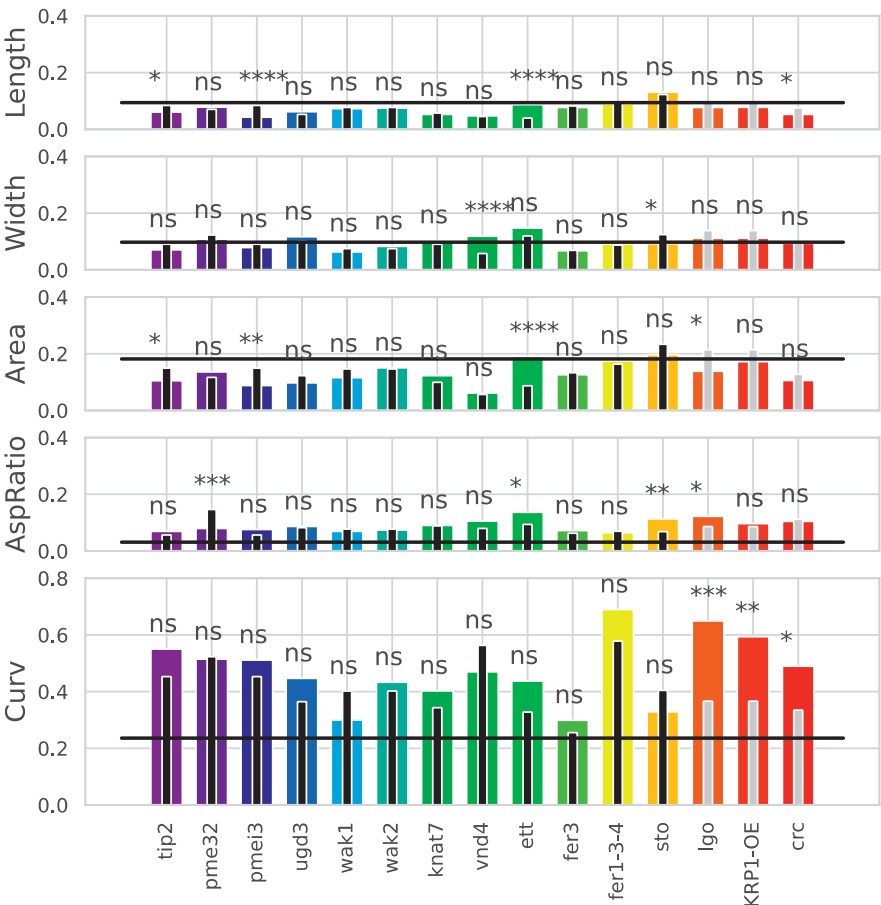

**Figure 5. Variability of mutants and corresponding wild-type phenotypes.** For each parameter, variability is shown as coloured bars for mutants and black (Col-0) or grey (Ler) vertical lines for the corresponding wild type. For each descriptor, the inter-experiment variability of the wild-type samples is represented by a horizontal black line. Significant differences were determined using Bartlett's test and indicated with asterisks: * if 0.005<p-value<=0.05, ** if 0.0005<p-value<=0.005, *** if 0.00005<p-value<=0.0005 and **** if p-value<=0.00005, and ns designates non-significant differences.

the wall-associated kinases (WAKs), a subfamily of receptor-like kinases that participate in cell wall integrity (CWI) sensing. The fact that *wak2*, but not *wak1*, exhibits a sepal phenotype compared to the wild type suggests that these two genes are not completely redundant. In accordance with this, in vitro studies show that WAK1 and WAK2 have a different affinity for demethylesterified pectins (Kohorn et al., 2009). WAK2 has been shown to regulate the transcriptional activity of invertase, opening the possibility of regulation of cell expansion via the control of turgor pressure. Interestingly, the expression of the *TIP2* (*DELTA TONOPLAST INTEGRAL PROTEIN 2*) gene, which encodes a tonoplast water channel, was shown to positively correlate with sepal width (Hartasánchez et al., 2023). Accordingly, we found here that sepals from the *tip2* mutant are narrower compared to those from wild-type plants.

PME activity has been associated with enhanced cell wall softening and cell expansion in shoot apical meristems and flower primordia (Peaucelle et al., 2011; Qiu et al., 2021). PME32 protein production has been shown to be completely abolished in the *pme32* mutant (Sénéchal, 2013). However, the mutation induces more expression of other PME genes and less expression of several genes encoding PME inhibitors (Sénéchal, 2013). Consequently, the total PME activity in this mutant is increased (Figure S1). As *pme32* mutant plants have more PME activity, longer and larger sepals are consistent with the idea that PMEs contribute to organ

elongation via a positive effect on growth. This mutant is another example of over-compensation, which leads to an effect opposite to what is expected.

Stomata have been shown to account for a third of the cells present on the abaxial epidermis of the sepal (Roeder et al., 2012). Sepals from the *sto* mutant, which has fewer stomata compared to the wild type, are longer than those of the wild type, and this increased length affects aspect ratio and area. This could indicate that in sepals, gas exchanges are not the limiting factor for development or that in our cultural conditions, this defect can be compensated for. Alternatively, we could imagine that stomata cells, being small, contribute to mechanical rigidity. Therefore, having less stomata would facilitate elongation by reducing mechanical resistance to sepal elongation.

The *CRABS CLAW* (*CRC*) gene controls gynoecium development, and mutations in this gene have been shown to affect gynoecium morphogenesis, in particular, producing shorter gynoecia (Bowman & Smyth, 1999). Our analyses show that sepals from the *crc* mutant are shorter, as expected if gynoecium growth was correlated to sepal growth. The *CRC* gene is only expressed in the gynoecium and nectaries (and not in sepals). Hence, the effect observed in sepals could be due to a mechanical effect of gynoecium size on sepal growth. Another gene involved in gynoecium length is *ETTIN* (*ETT*), with the *ett* mutant showing a reduced gynoecium length. *ETT* has a broad expression and is involved

in many biological functions, including apico-basal and abaxial-adaxial patterning (Sessions & Zambryski, 1995). As observed for the *crc* mutant, the reduction of gynoecium length correlates with the reduction of sepal length, supporting a possible role of gynoecium elongation in sepal length. However, *ETT* is expressed in almost all organs, including sepals, so a direct role of ETT in sepal development cannot be discarded.

Lastly, *VND4* expression has been shown to be negatively correlated with sepal width in the wild type (Hartasánchez et al., 2023). We found increased *VND4* expression in the mutant analysed (Figure S1). We therefore expect a decrease in sepal width in this mutant, and it is indeed what we observed (Table 2). VND4 belongs to the secondary wall NAC master switches (Zhong & Ye, 2015), capable of turning on the secondary cell wall biosynthetic program, a process which rigidifies the tissues (Zhong & Ye, 2015) and could lead to growth arrest.

### 4.3. Correlation between length and curvature

In our previous work (Hartasánchez et al., 2023), we found a negative correlation between sepal size and curvature among sepals from wild-type plants. In the mutants analysed here, although we observe smaller sepals in lgo, *KRP1*, *ett*, *fer1-3-4* and *crc* mutants, the curvature is only increased in *ett* (in line with previous observations for the wild type), whereas it is decreased in *KRP1-OE* and *crc*, and is not affected in *lgo* and *fer1-3-4*. Similarly, larger sepals are observed in *pme32*, *sto* and *fer3* with no effect on curvature for *pme32*, *sto*, and only a negative effect for *fer3*. Therefore, the correlation between sepal curvature and size does not hold for the mutants analysed (also visible in Figure 3).

### 4.4. Variability of sepal morphology

Among sepals from wild-type plants, we observe significant variability in sepal morphology, although Col-0 (or Ler) plants from the same batch of seeds have been used, excluding genetic heterogeneity. This variability could originate from small and local variations in culture conditions despitetight control of growth conditions. In addition, gene expression is a stochastic process, and at the cellular level, we can distinguish extrinsic noise, which affects the expression of all the genes equally, and intrinsic noise, which differs from one gene to the other and is due to the inherent stochasticity of transcription and translation (Araújo et al., 2017). It is likely that all these sources of variability contribute to the morphological differences observed across wild-type sepals. We also find a significant correlation between length and area variabilities, an anti-correlation between the curvature and its variability, and a correlation between the aspect ratio and its variability. The reasons for these observations probably result from interactions among the mechanisms regulating sepal morphology, although the origin and nature of these interactions remain to be determined.

Interestingly, we observed a decrease in variability for length and area in the *pmei3* mutant and aspect ratio for *pme32*. This observation supports a role for pectin methylesterification in the reproducibility of sepal morphology. On the opposite, we found several mutants exhibiting more variability on at least one parameter of sepal morphology: *sto*, *ett*, *lgo* and *KRP1-OE*. Among those, three mutations affect cell type, suggesting that an equilibrated proportion of different cell types on the abaxial epidermis is important for the reproducibility of sepal geometry. A recent publication found that sepal shape variability is robust to cell size heterogeneity (Trinh et al., 2024). However, their analysis did not take curvature

into account since they flattened sepals. Here, we show that both *lgo* and *KRP1-OE* plants have sepals with increased shape variability (aspect ratio and curvature). The aspect ratio measured here is using curvilinear length and width, which may differ significantly from the length and width of flattened sepals. This points to the importance of including the curvature when studying sepal shape.

The increased variability of sepal morphology in the *ett* mutant is reminiscent of variability in gynoecium morphology in this mutant, suggesting a role for auxin signalling in the reproducibility of flower organ morphology. The fact that a certain level of variability is observed among the different wild-type plants and that several mutants have increased variability suggests that selection allows an intermediate level of variability but has ensured that mechanisms are at work to keep a minimum of phenotypic robustness.

**Supplementary material.** The supplementary material for this article can be found at http://doi.org/10.1017/qpb.2025.5.

**Data availability statement.** All the data that were used for the statistical analysis in this study (original images, segmented images, as well as sepal measurements) are openly available as a published dataset from the French national data platform recherche.data.gouv.fr at https://doi.org/10.57745/LTTTBK. No custom code or scripts were developed for the analysis; all statistical tests were performed using the scipy.stats module, principal component analysis (PCA) was conducted using sklearn.decomposition, and visualisations were created using the matplotlib Python package.

## Acknowledgements

We thank Adrienne Roeder, Bruce Kohorn, Jérôme Pelloux and Joan Renard for sharing seeds, Annick Dubois and Thierry Gaude for reading the manuscript and PLATIM for providing microscope facilities.

**Author contributions.** Virginie Battu and Annamaria Kiss have equal contributions. DH, AB and FM conceived and designed the study. VB, ADV and FS conducted data gathering. VB, AK and CM performed statistical analyses. AK, DH, AB and FM wrote the article.

**Funding statement.** This research received no specific grant from any funding agency, commercial or not-for-profit sectors.

**Competing interest.** The authors declare no competing interests exist.

**Open peer review.** To view the open peer review materials for this article, please visit http://doi.org/10.1017/qpb.2025.5.

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
