## [Reviewer Report]

Battu, Kiss, and co-workers investigate a possible genetic framework for robust organ development using Arabidopsis thaliana sepal as a model. They employed morphological 3D quantification and statistical analysis with a series of mutant lines to extract key parameters and their interactions relevant to sepal morphogenesis. This experiment identified that, e.g., sepal shape is more robust than sepal size, which would be hard to show through conventional methods. The content overall fits well with Quantitative Plant Biology. While I appreciate the significance of the findings, certain aspects require attention.

Major comments:

1) Correlation analysis of sepal morphological parameters in Col-0 was previously reported from the authors’ group, showing that sepal size is anti-correlated with curvature (Hartasánchez et al., 2023). The current study extends this approach using 15 mutant lines. To make this point clearer to the readers, I recommend adding some more explanation of the previous work and the already obtained results when framing a research question in the Introduction section.

2) The authors described the molecular function of the focused genes to link it with the observed morphological changes in the Discussion section. This would be more convincing if additional cellular-level quantitative data (or representative cellular-level images) are shown to fill the gap between molecular function and organ morphology. Alternatively, please summarize the description of the molecular function more briefly to highlight what the authors found in this study.

3) I question the conclusion presented in lines 623-629, which states how the robustness of the sepal morphology is established. The authors measured the morphological parameters under the uniform culture condition and detected, for example, a decrease in variability for length and area within the pmei3 mutants compared with WT. It is unclear to me what the pmei3 mutant is robust against. In my opinion, pectin methylesterification helps sustain the “reproducibility” of the sepal morphology when cultured under one condition, however, it remains elusive whether this is relevant to “robustness” against some perturbation in WT. It would be better to discriminate between “reproducibility” and “robustness” throughout the manuscript to avoid confusion.

Minor comments:

4) PME activity increased in the pme32-2 mutants (Figure S1C). How do the author explain that?

5) Please add a more detailed explanation for lines 349-351 to increase readability.

6) Figure 4C was not cited throughout the manuscript.

7) Data visualization in Figure 5 is interesting and convincing for me. I like this panel.

---

## [Reviewer Report]

How robust the three-dimensional structure of the organ has not so far been analyzed in detail. The authors addressed this issue using sepals of Arabidopsis thaliana as a model. The sepals of 15 mutants, including wild type (WT), were analyzed with a newly developed pipeline. Then, the results were compared to investigate which geographic parameter is more robust during sepal morphogenesis. The results suggest that sepal curvature is an important parameter for variations in sepal morphology within genotypes. The authors also examined the robustness of sepal morphology in WT and showed that the shape of sepals was more robust than their size.

Despite the potentially interesting findings, there are major and minor problems regarding experimental methods and interpretation of results. In this current state, this manuscript is quite descriptive and lacks a clear mechanistic insight. For example, the authors measured several parameters (width, length, area, curve) of sepals in mutants and discussed the relationship between each parameter and epidermal cell shape. However, the organ shape is not determined by epidermal cells alone, but the division pattern and timing of the cells inside should also be considered. Indeed, recent studies have revealed the importance of cell proliferation position in the organ. The authors ignored those factors in the current manuscript and should also measure cell size and cell number within sepals at least. Therefore, many conclusions on differences in sepal shape cannot be drawn from such limited data.

Minor comments

L1; The title should be revised. In particular, the word “regulation” needs to be revised. The current title is “A 3D perspective on the regulation of Arabidopsis sepal~”. However, in the current manuscript, no specific data or discussion on the regulation of sepal morphology.

Methods; At what developmental stage were the sepal samples collected for each experiment (e.g., RNA extraction and sepal shape analysis)?

L216; Please provide more information on “high module membership and high gene significance”.

L307; The authors stated that “Length and width are also positively correlated suggesting that shape is under selective constraint.”. What does this “selective constraint” mean? Evolutionally constraint? Please clarify this point.

L513; The authors discussed expression levels of PMEs in pme mutant plants and found that the elevated expression levels were responsible for the sepal phenotypes. How about sepal phenotypes of PME overexpressors? If the phenotypes were the same, the discussion on PMEs would be more solid.

L537; The authors stated, “Surprisingly, sepals from the sto mutant are longer than those of the wild type~”. To test the generality of this result, the leaves of the same mutant can be observed. As both sepals and leaves are the same lateral organs, and they exchange gases, this should provide good data to test the validity of the present discussion.

---

## [Editor Report]

Dear authors,

In this manuscript, Battu and co-workers are investigating a possible genetic framework for robust plant organ development using the sepals of Arabidopsis thaliana as a model organ. They conducted morphological 3D quantification and statistical analysis with a series of mutant lines to extract key parameters and their interactions relevant to sepal morphogenesis. They found that, e.g., sepal shape is more robust than sepal size, which would be hard to distinguish through conventional methods. 

Now we have received the comments of two reviewers. While they found that the overall content of the manuscript fits well with the scope of Quantitative Plant Biology, and that the findings are significant to the field, they also found that certain aspects require further attention.

More specifically, and in brief, since the present manuscript is a follow up work of previous research, the authors are required (1) to add more explanation of the previous work and the obtained results when framing a research question in the Introduction. (2) While describing the molecular function of the selected genes to link it with the observed morphological changes in the Discussion, additional cellular-level quantitative data (or representative cellular-level images) are required to convincingly fill the gap between molecular function and organ morphology. (3) It would be better to discriminate between “reproducibility” and “robustness” throughout the manuscript to avoid confusion. In addition to the above issues, the reviewers have raised additional Major and minor comments for the authors to consider. (4) L1; The title should be revised. In particular, the word “regulation” needs to be revised. The current title is “A 3D perspective on the regulation of Arabidopsis sepal~”. However, in the current manuscript, no specific data or discussion on the regulation of sepal morphology.(5) Methods; At which developmental stage were the sepal samples collected for each experiment (e.g., RNA extraction and sepal shape analysis).

Based on the reviewers reports and my personal evaluation, as a minimum requirement I would like to invite the authors to revise the manuscript based on the reviewers comments to come up with a thoroughly revised version that should be resubmitted for consideration in QPB.

In the revised manuscript, please mark in Red-Inc the changes included, and please submit separately a point-by-point response letter indicating all the modifications in detail.

I am looking forward to receiving your revised manuscript.

ALI FERJANI

Associate Editor of QPB

---

## [Reviewer Report]

The authors sufficiently responded to my previous comments. I have now supported the publication of this manuscript.

---

## [Reviewer Report]

Although the authors responded well to the reviewer’s comments, some minor details need to be clarified before this work can be accepted for publication.

Minor comments

L111: Please provide details of the plant material section (e.g., soil, watering, growth chamber/greenhouse, etc.)

Figure 1; Figure 1 shows that the thickness of sepals differs among mutants. It would be a good idea for the authors to discuss the relationship between the thickness and other parameters because the thickness is one of the important parameters in 3D morphology. Alternatively, it may be worthwhile to provide thickness data as supplemental data.

---

## [Editor Report]

Dear Dr. Monéger,

Thank you for submitting your revised manuscript to QPB.

Now your revised manuscript has been evaluated again by the original reviewers, and we have reached the decision of minor revision.

One of the reviewers found that although the authors responded well to his/her comments, some minor details need to be clarified before this work can be accepted for publication.

More specifically:

(1) L111: Please provide details of the plant material section (e.g., soil, watering, growth chamber/greenhouse, etc.)

(2) Figure 1; Figure 1 shows that the thickness of sepals differs among mutants. It would be a good idea for the authors to discuss the relationship between the thickness and other parameters because the thickness is one of the important parameters in 3D morphology. Alternatively, it may be worthwhile to provide thickness data as supplemental data.

As you may have appreciated, both comments can be easily addressed by text addition/ and or modification and quick measurement of thickness.

I am looking forward to receiving the revised manuscript.

Thank you again for submitting your nice work to QPB

---

## [Editor Report]

Dear authors,

QPB-2024-0061.R2 entitled “A 3D morpho-space of sepal geometry reveals the importance of organ curvature” which you submitted to Quantitative Plant Biology, has been reviewed. We found that the two remaining minor points have been properly addressed. Based on this, the reviewers and myself are happy to recommend the manuscript for publication as is. Congratulations!

Thank you again for submitting your work to QPB

Ali FERJANI

Associate Editor, QPB